# MetaAnchor: Learning to Detect Objects with Customized Anchors

**Tong Yang**[*†]     **Xiangyu Zhang**[*]     **Zeming Li**[*]     **Wenqiang Zhang**[†]     **Jian Sun**[*]

[*] **Megvii Inc (Face++)**        [†] **Fudan University**

{yangtong,zhangxiangyu,lizeming,sunjian}@megvii.com    wqzhang@fudan.edu.cn

## Abstract

We propose a novel and flexible anchor mechanism named MetaAnchor for object detection frameworks. Unlike many previous detectors model anchors via a predefined manner, in MetaAnchor anchor functions could be dynamically generated from the arbitrary customized prior boxes. Taking advantage of weight prediction, MetaAnchor is able to work with most of the anchor-based object detection systems such as RetinaNet. Compared with the predefined anchor scheme, we empirically find that MetaAnchor is more robust to anchor settings and bounding box distributions; in addition, it also shows the potential on transfer tasks. Our experiment on COCO detection task shows that MetaAnchor consistently outperforms the counterparts in various scenarios.

## 1 Introduction

The last few years have seen the success of deep neural networks in object detection task [5, 39, 9, 12, 8, 32, 16, 2]. In practice, object detection often requires to generate a set of bounding boxes along with their classification labels associated with each object in the given image. However, it is nontrivial for convolutional neural networks (CNNs) to directly predict an orderless set of arbitrary cardinality[1]. One widely-used workaround is to introduce *anchor*, which employs the thought of divide-and-conquer and has been successfully demonstrated in the state-of-the-art detection frameworks [39, 32, 25, 30, 31, 11, 22, 23, 2]. In short, anchor method suggests dividing the box space (including position, size, class, etc.) into discrete bins (not necessarily disjoint) and generating each object box via the *anchor function* defined in the corresponding bin. Denote $\mathbf{x}$ as the feature extracted from the input image, then anchor function for $i$-th bin could be formulated as follows:

$$\mathcal{F}_{b_i}(\mathbf{x}; \theta_i) = \left( \mathcal{F}_{b_i}^{cls}(\mathbf{x}; \theta_i^{cls}), \mathcal{F}_{b_i}^{reg}(\mathbf{x}; \theta_i^{reg}) \right) \tag{1}$$

where $b_i \in \mathcal{B}$ is the *prior* (also named *anchor box* in [32]), which describes the common properties of object boxes associated with $i$-th bin (e.g. averaged position/size and classification label); while $\mathcal{F}_{b_i}^{cls}(\cdot)$ discriminates whether there exists an object box associated with the $i$-th bin, and $\mathcal{F}_{b_i}^{reg}(\cdot)$ regresses the relative location of the object box (if any) to the prior $b_i$; $\theta_i$ represents the parameters for the anchor function.

To model anchors with deep neural networks, one straight-forward strategy is via *enumeration*, which is adopted by most of the previous work [32, 39, 25, 30, 31, 23, 11, 22]. First, a number of predefined priors (or anchor boxes) $\mathcal{B}$ is chosen by handcraft [32] or statistical methods like clustering [39, 31]. Then for each $b_i \in \mathcal{B}$ the anchor function $\mathcal{F}_{b_i}$ is usually implemented by one or a few neural network layers respectively. Weights for different anchor functions are independent or partially shared. Obviously in this framework anchor strategies (i.e. anchor box choices and the definition of

corresponding anchor functions) are fixed in both training and inference. In addition, the number of available anchors is limited by the predefined $\mathcal{B}$.

In this paper, we propose a flexible alternative to model anchors: instead of enumerating every possible bounding box prior $b_i$ and modeling the corresponding anchor functions respectively, in our framework anchor functions are dynamically generated from $b_i$. It is done by introducing a novel *MetaAnchor* module which is defined as follows:

$$\mathcal{F}_{b_i} = \mathcal{G}\left(b_i; w\right) \tag{2}$$

where $\mathcal{G}(\cdot)$ is called *anchor function generator* which maps any bounding box prior $b_i$ to the corresponding anchor function $\mathcal{F}_{b_i}$; and $w$ represents the parameters. Note that in MetaAnchor the prior set $\mathcal{B}$ is not necessarily predefined; instead, it works as a **customized** manner – during inference, users could specify any anchor boxes, generate the corresponding anchor functions and use the latter to predict object boxes. In Sec. 3, we present that with *weight prediction* mechanism [10, 18] anchor function generator could be elegantly implemented and embedded into existing object detection frameworks for joint optimization.

In conclusion, compared with traditional predefined anchor strategies, we find our proposed MetaAnchor has the following potential benefits (detailed experiments are present in Sec. 4):

- **MetaAnchor is more robust to anchor settings and bounding box distributions.** In traditional approaches, the predefined anchor box set $\mathcal{B}$ often needs careful design – too few anchors may be insufficient to cover rare boxes, or result in coarse predictions; however, more anchors usually imply more parameters, which may suffer from overfitting. In addition, many traditional strategies use independent weights to model different anchor functions, so it is very likely for the anchors associated with few ground truth object boxes in training to produce poor results. In contrast, for MetaAnchor anchor boxes of any shape could be randomly sampled during training so as to cover different kinds of object boxes, meanwhile, the number of parameters keeps constant. Furthermore, according to Equ. 2 different anchor functions are generated from the same weights $w$, thus all the training data are able to contribute to all the model parameters, which implies more robustness to the distribution of the training boxes.

- **MetaAnchor helps to bridge the bounding box distribution gap between datasets.** In traditional framework, anchor boxes $\mathcal{B}$ are predefined and keep unchanged for both training and test, which could be suboptimal for either dataset if their bounding box distributions are different. While in MetaAnchor, anchors could be flexibly customized to adapt the target dataset (for example, via grid search) without retraining the whole detector.

## 2   Related Work

**Anchor methodology in object detection.**     Anchors (maybe called with other names, e.g. "default boxes" in [25], "priors" in [39] or "grid cells" in [30]) are employed in most of the state-of-the-art detection systems [39, 32, 22, 23, 25, 7, 11, 2, 31, 21, 35, 15]. The essential of anchors includes position, size, class label or others. Currently most of the detectors model anchors via enumeration, i.e. predefining a number of anchor boxes with all kinds of positions, sizes and class labels, which leads to the following issues. First, anchor boxes need careful design, e.g. via clustering [31], which is especially critical on specific detection tasks such as anchor-based face [40, 45, 28, 36, 43] and pedestrian [41, 3, 44, 26] detections. Specially, some papers suggest multi-scale anchors [25, 22, 23] to handle different sizes of objects. Second, predefined anchor functions may cause too many parameters. A lot of work addresses the issue by weight sharing. For example, in contrast to earlier work like [5, 30], detectors like [32, 25, 31] and their follow-ups [7, 22, 2, 11, 23] employ *translation-invariant anchors* produced by fully-convolutional network, which could share parameters across different positions. Two-stage frameworks such as [32, 2] share weights across various classes. And [23] shares weights for multiple detection heads. In comparison, our approach is free of the issues, as anchor functions are customized and generated dynamically.

**Weight prediction.**     Weight prediction means a mechanism in neural networks where weights are predicted by another structure rather than directly learned, which is mainly used in the fields of *learning to learn* [10, 1, 42], few/zero-shot learning [4, 42] and transfer learning [27]. For object

detection there are a few related works, for example, [15] proposes to predict mask weights from box weights. There are mainly two differences from ours: first, in our MetaAnchor the purpose of weight prediction is to generate anchor functions, while in [15] it is used for domain adaption (from object box to segmentation mask); second, in our work weights are generated almost "from scratch", while in [15] the source is the learned box weights.

# 3 Approach

## 3.1 Anchor Function Generator

In MetaAnchor framework, *anchor function* is dynamically generated from the customized box prior (or anchor box) $b_i$ rather than fixed function associated with predefined anchor box. So, *anchor function generator* $\mathcal{G}(\cdot)$ (see Equ. 2), which maps $b_i$ to the corresponding anchor function $\mathcal{F}_{b_i}$, plays a key role in the framework. In order to model $\mathcal{G}(\cdot)$ with neural work, inspired by [15, 10], first we assume that for different $b_i$ anchor functions $\mathcal{F}_{b_i}$ share the same formulation $\mathcal{F}(\cdot)$ but have different parameters, which means:

$$\mathcal{F}_{b_i}(\mathbf{x}; \theta_i) = \mathcal{F}(\mathbf{x}; \theta_{b_i}) \tag{3}$$

Then, since each anchor function is distinguished only by its parameters $\theta_{b_i}$, anchor function generator could be formulated to predict $\theta_{b_i}$ as follows:

$$\begin{aligned}
\theta_{b_i} &= \mathcal{G}(b_i; w) \\
&= \theta^* + \mathcal{R}(b_i; w)
\end{aligned} \tag{4}$$

where $\theta^*$ stands for the shared parameters (independent to $b_i$ and also learnable), and the residual term $\mathcal{R}(b_i, w)$ depends on anchor box $b_i$.

In the paper we implement $\mathcal{R}(\cdot)$ with a simple two-layer network:

$$\mathcal{R}(b_i, w) = W_2 \sigma (W_1 b_i) \tag{5}$$

Here, $W_1$ and $W_2$ are the learnable parameters and $\sigma(\cdot)$ is the activation function (i.e. ReLU in our work). Denote the number of hidden neurons by $m$. In practice $m$ is usually much smaller than the dimension of $\theta_{b_i}$, which causes the weights predicted by $\mathcal{R}(\cdot)$ lie in a significantly low-rank subspace. That is why we formulate $\mathcal{G}(\cdot)$ as a residual form in Equ 4 rather than directly use $\mathcal{R}(\cdot)$. We also survey more complex designs for $\mathcal{G}(\cdot)$, however, which results in comparable benchmarking results.

In addition, we introduce a data-dependent variant of anchor function generator, which takes the input feature $\mathbf{x}$ into the formulation:

$$\begin{aligned}
\theta_{b_i} &= \mathcal{G}(b_i; \mathbf{x}, w) \\
&= \theta^* + W_2 \sigma (W_{11} b_i + W_{12} r(\mathbf{x}))
\end{aligned} \tag{6}$$

where $r(\cdot)$ is used to reduce the dimension of the feature $\mathbf{x}$; we empirically find that for convolutional feature $\mathbf{x}$, using *global averaged pooling* [13, 38] operation for $r(\cdot)$ usually produces good results.

## 3.2 Architecture Details

Theoretically MetaAnchor could work with most of the existing anchor-based object detection frameworks [32, 25, 30, 31, 23, 11, 22, 19, 20, 2]. Among them, for the two-stage detectors [32, 2, 22, 11, 19] anchors are usually used to model "objectness" and generate box proposals, while fine results are predicted by RCNN-like modules [9, 8] in the second stage. We try to use MetaAnchor in these frameworks and observe some improvements on the box proposals (e.g. improved recalls), however, it seems no use to the final predictions, whose quality we believe is mainly determined by the second stage. Therefore, in the paper we mainly study the case of single-stage detectors [30, 25, 31, 23].

We choose the state-of-the-art single-stage detector *RetinaNet* [23] to apply MetaAnchor for instance. Note that our methodology is also applicable to other single-stage frameworks such as [31, 25, 7, 35]. Fig 1(a) gives the overview of RetinaNet. In short, 5 levels of features $\{P_l | l \in \{3, 4, 5, 6, 7\}\}$ are extracted from a "U-shaped" backbone network, where $P_3$ stands for the finest feature map (i.e. with largest resolution) and $P_7$ is the coarsest. For each level of feature, a subnet named "detection head" in Fig 1 is attached to generate detection results. Anchor functions are defined at the tail of

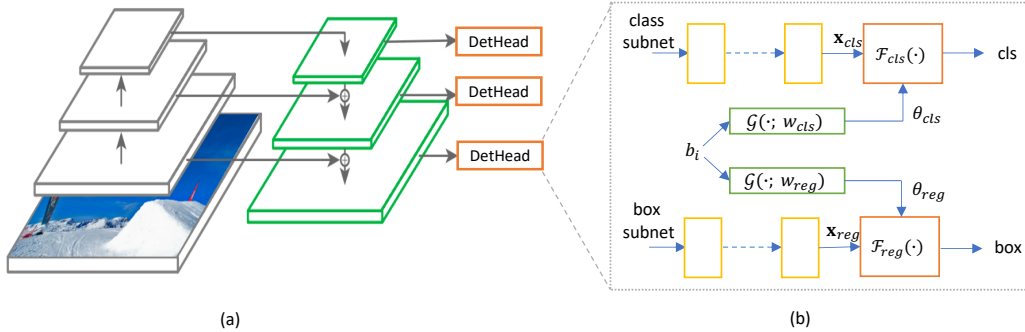

(a)                                                                                                    (b)

Figure 1: Illustration to applying MetaAnchor on *RetinaNet* [23]. (a) RetinaNet overview. (b) Detection heads in RetinaNet equipped with MetaAnchor. $\mathcal{F}_{cls}(\cdot)$ and $\mathcal{F}_{reg}(\cdot)$ compose the *anchor function* (defined in Equ 1), which are implemented by a convolutional layer respectively here. $\mathcal{G}(\cdot, w_{cls})$ and $\mathcal{G}(\cdot, w_{reg})$ are *anchor function generators* defined in Equ 4 (or Equ 6). $b_i$ is the customized box prior (or called anchor box); and "cls" and "reg" represent the prediction results associated to $b_i$.

each detection head. Referring to the settings in [23], anchor functions are implemented by a $3 \times 3$ convolutional layer; and for each detection head, there are $3 \times 3 \times 80$ types of anchor boxes (3 scales, 3 aspect ratios and 80 classes) are predefined. Thus for each anchor function, there should be 720 filters for the classification term and 36 filters for the regression term ($3 \times 3 \times 4$, as regression term is class-agnostic).

In order to apply MetaAnchor, we need to redesign the original anchor functions so that their parameters are generated from the customized anchor box $b_i$. First of all, we consider how to encode $b_i$. According to the definition in Sec. 1, $b_i$ should be a vector which includes the information such as position, size and class label. In RetinaNet, thanks to the fully-convolutional structure, position could be naturally encoded by the coordinate of feature maps thus no need to be involved in $b_i$. As for class label, there are two alternatives: A) directly encode it in $b_i$, or B) let $\mathcal{G}(\cdot)$ predict weights for each class respectively. We empirically find that Option B is easier to optimize and usually results in better performance than Option A. So, in our experiment $b_i$ is mainly related to anchor size. Motivated by the bounding box encoding method introduced in [9, 32], $b_i$ is represented as follows:

$$b_i = \left( \log \frac{ah_i}{AH}, \log \frac{aw_i}{AW} \right) \qquad (7)$$

where $ah_i$ and $aw_i$ are the height and width of the corresponding anchor box; and $(AH, AW)$ is the size of "standard anchor box", which is used as a normalization term. We also survey a few other alternatives, for example, using the scale and aspect ratio to represent the size of anchor boxes, which results in comparable results with that of Equ. 7.

Fig 1(b) illustrates the usage of MetaAnchor in each detection head of RetinaNet. In the original design [23], the classification and box regression parts of anchor functions are attached to separated feature maps ($\mathbf{x}_{cls}$ and $\mathbf{x}_{reg}$) respectively; so in MetaAnchor, we also use two independent *anchor function generators* $\mathcal{G}(\cdot, w_{cls})$ and $\mathcal{G}(\cdot, w_{reg})$ to predict their weights respectively. The design of $\mathcal{G}(\cdot)$ follows Equ. 4 (data-independent variant) or Equ. 6 (data-dependent variant), in which the number of hidden neurons $m$ is set to 128. In addition, recall that in MetaAnchor anchor functions are dynamically derived from $b_i$ rather than predefined by enumeration; so, the number of filters for $\mathcal{F}_{cls}(\cdot)$ reduces to 80 (80 classes, for example) and 4 for $\mathcal{F}_{reg}(\cdot)$.

It is also worth noting that in RetinaNet [23] corresponding layers in all levels of detection heads share the same weights, even including the last layers which stand for anchor functions. However, the definitions of anchors differ from layer to layer: for example, in $l$-th level suppose an anchor function associated to the anchor box of size $(ah, aw)$; while in $(l+1)$-th level (with 50% smaller resolution), the same anchor function should detect with 2x larger anchor box, i.e. $(2ah, 2aw)$. So, in order to keep consistent with the original design, in MetaAnchor we use the same anchor generator function $\mathcal{G}(\cdot, w_{cls})$ and $\mathcal{G}(\cdot, w_{reg})$ for each level of detection head; while the "standard boxes" $(AH, AW)$ in Equ. 7 are different between levels: suppose the standard box size in $l$-th level is $(AH_l, AW_l)$, then for $(l+1)$-th level we set $(AH_{l+1}, AW_{l+1}) = (2AH_l, 2AW_l)$. In our experiment, the size of

Table 1: Anchor box configurations

| # of Anchors | Scales [2] | Aspect Ratios | $(AH, AW)$ |
|---|---|---|---|
| $3 \times 3$ | $\{2^{k/3}|k < 3\}$ | $\{1/2, 1, 2\}$ | $(44, 44)$ |
| $5 \times 5$ | $\{2^{k/5}|k < 5\}$ | $\{1/3, 1/2, 1, 2, 3\}$ | $(45, 47)$ |
| $7 \times 7$ | $\{2^{k/7}|k < 7\}$ | $\{1/4, 1/3, 1/2, 1, 2, 3, 4\}$ | $(48, 50)$ |
| $9 \times 9$ | $\{2^{k/9}|k < 9\}$ | $\{1/5, 1/4, 1/3, 1/2, 1, 2, 3, 4, 5\}$ | $(53, 53)$ |

standard box in the lowest level (i.e. $P_3$, which has the largest resolution) is set to the average of all the anchor box sizes (shown in the last column in Table 1).

## 4  Experiment

In this section we mainly evaluate our proposed *MetaAnchor* on COCO object detection task [24]. The basic detection framework is *RetinaNet* [23] as introduced in 3.2, whose backbone feature extractor we use is ResNet-50 [13] pretrained on ImageNet classification dataset [34]. For MetaAnchor, we use the data-independent variant of anchor function generator (Equ. 4) unless specially mentioned. MetaAnchor subnets are jointly optimized with the backbone detector during training. We do not use *Batch Normalization* [17] in MetaAnchor.

**Dataset.**   Following the common practice [23] in COCO detection task, for training we use two different dataset splits: *COCO-all* and *COCO-mini*; while for test, all results are evaluated on the *minival* set which contains 5000 images. COCO-all includes all the images in the original training and validation sets excluding *minival* images, while COCO-mini is a subset of around 20000 images. Results are mainly evaluated with COCO standard metrics such as mmAP.

**Training and evaluation configurations.**   For fair comparison, we follow most of the settings in [23] (image size, learning rate, etc.) for all the experiments, except for a few differences as follows. In [23], $3 \times 3$ anchor boxes (i.e. 3 scales and 3 aspect ratios) are predefined for each level of detection head. In the paper, more anchor boxes are employed in some experiments. Table 1 lists the anchor box configurations for feature level $P_3$, where the $3 \times 3$ case is identical to that in [23]. Settings for other feature levels could also be derived (see Sec. 3.2). As for MetaAnchor, since predefined anchors are not needed, we suggest to use the strategy as follows. In training, first we select a sort of anchor box configuration from Table 1 (e.g. $5 \times 5$), then generate 25 $b_i$s according to Equ. 7; for each iteration, we randomly augment each $b_i$ within $\pm 0.5$, calculating the corresponding ground truth and use them to optimize. We call the methodology "training with $5 \times 5$ anchors". While in test, $b_i$s are also set by a certain anchor box configuration without augmentation (not necessary the same as used in training). We argue that with that training/inference scheme, it is possible to make direct comparisons between MetaAnchor and the counterpart baselines.

In the following subsections, first we study the performances of MetaAnchor by a series of controlled experiments on COCO-mini. Then we report the fully-equipped results on COCO-full dataset.

### 4.1  Ablation Study

#### 4.1.1  Comparison with RetinaNet baselines

Table 2 compares the performances of MetaAnchor and RetinaNet baseline on COCO-mini dataset. Here we use the same anchor box settings for training and test. In the column "Threshold" $t_1/t_2$ means the intersection-over-union (IoU) thresholds for positive/negative anchor boxes respectively in training (the detailed definition are introduced in [32, 23]).

To analyze, first we compare the rows with the threshold of 0.5/0.4. It is clear that MetaAnchor outperforms the counterpart baselines on each of anchor configurations and evaluation metrics, for instance, $0.2 \sim 0.8\%$ increase for mmAP and $0.8 \sim 1.5\%$ for $AP_{50}$. We suppose the improvements may come from two aspects: first, in MetaAnchor the sizes of anchor boxes could be augmented and

Table 2: Comparison of RetinaNets with/without MetaAnchor.

| Threshold | # of Anchors | Baseline (%) | | | MetaAnchor (%) | | |
|---|---|---|---|---|---|---|---|
| | | mmAP | $AP_{50}$ | $AP_{75}$ | mmAP | $AP_{50}$ | $AP_{75}$ |
| 0.5/0.4 | $3 \times 3$ | 26.5 | 43.1 | 27.6 | **26.9** | 44.2 | 28.2 |
| 0.5/0.4 | $5 \times 5$ | 26.9 | 43.7 | 28.1 | **27.1** | 44.5 | 28.1 |
| 0.5/0.4 | $7 \times 7$ | 26.4 | 43.0 | 27.7 | **27.2** | 44.4 | 28.5 |
| 0.5/0.4 | $9 \times 9$ | 26.3 | 42.8 | 27.5 | **27.1** | 44.3 | 28.2 |
| 0.6/0.5 | $3 \times 3$ | 25.7 | 41.1 | 27.3 | **26.0** | 42.0 | 27.2 |
| 0.6/0.5 | $5 \times 5$ | 26.1 | 41.4 | 27.8 | **27.3** | 44.2 | 28.8 |
| 0.6/0.5 | $7 \times 7$ | 26.2 | 41.3 | 27.9 | **27.0** | 43.1 | 28.3 |
| 0.6/0.5 | $9 \times 9$ | 26.1 | 41.0 | 27.9 | **27.4** | 43.7 | 29.2 |

Table 3: Comparison of various anchors in inference (mmAP, %)

| Training | Inference | | | | |
|---|---|---|---|---|---|
| # of Anchors | $3 \times 3$ | $5 \times 5$ | $7 \times 7$ | $9 \times 9$ | search |
| $3 \times 3$ | 26.0 | 26.6 | 26.8 | 26.7 | **27.0** |
| $5 \times 5$ | 26.7 | 27.3 | 27.5 | 27.5 | **27.7** |
| $7 \times 7$ | 26.1 | 26.9 | 27.0 | 27.1 | **27.3** |
| $9 \times 9$ | 26.3 | 27.2 | 27.4 | 27.4 | **27.6** |

make the anchor functions to generate a wider range of predictions, which may enhance the model capability (especially important for the case with smaller number of anchors, e.g. $3 \times 3$); second, rather than predefined anchor functions with independent parameters, MetaAnchor allows all the training boxes to contribute to the shared generators, which seems beneficial to the robustness over the different configurations or object box distributions.

For further investigating, we try using stricter IoU threshold (0.6/0.5) for training to encourage more precise anchor box association, however, statistically there are fewer chances for each anchor to be assigned with a positive ground truth. Results are also presented in Table 2. We find results of all the baseline models suffer from significantly drops especially on $AP_{50}$, which implies the degradation of anchor functions; furthermore, simply increasing the number of anchors works little on the performance. For MetaAnchor, in contrast, 3 out of 4 configurations are less affected (for the case of $9 \times 9$ anchors even 0.3% improved mmAP are obtained). The only exception is the $3 \times 3$ case; however, according to Table 3 we believe the degradation is mainly because of too few anchor boxes for inference rather than poor training. So, the comparison supports our hypothesis: MetaAnchor helps to use training samples in a more efficient and robust way.

### 4.1.2 Comparison of various anchor configurations in inference

Unlike the traditional fixed or predefined anchor strategy, one of the major benefits of MetaAnchor is able to use flexible anchor scheme during inference time. Table 3 compares a variety of anchor box configurations (refer to Table 1; note that the normalization coefficient $(AH, AW)$ should be consistent with what used in training) for inference along with their scores on COCO-mini. For each experiment IoU threshold in training is set to 0.6/0.5. From the results we find that more anchor boxes in inference usually produce higher performances, for instance, results of $9 \times 9$ inference anchors are $0.7 \sim 1.1\%$ better than that of $3 \times 3$ for a variety of training configurations.

Table 3 also implies that the improvements are quickly saturated with the increase of anchor boxes, e.g. $\geq 7 \times 7$ anchors only bring minor improvements, which is also observed in Table 2. We revisit the anchor configurations in Table 1 and find $7 \times 7$ and $9 \times 9$ cases tend to involve too "dense" anchor boxes, thus predicting highly overlapped results which might contribute little to the final performance. Inspired by the phenomenon, we come up with an inference approach via greedy search: each step we randomly select one anchor box $b_i$, generate the predictions and evaluate the combined results with the previous step (performed on a subset of training data); if the score improves, we update the current predictions with the combined results, otherwise discard the predictions in the current step. Final anchor configuration is obtained after a few steps. Improved results are shown in the last column (named "search") of Table 3.

Table 4: Comparison in the scenarios of different training/test distributions (mmAP, %)

| # of Anchors | Baseline (all) | MetaAnchor (all) | Baseline (drop) | MetaAnchor (drop) |
|:---:|:---:|:---:|:---:|:---:|
| $3 \times 3$ | 26.5 | **26.9** | 21.2 | **22.2** |
| $5 \times 5$ | 26.9 | **27.1** | 20.8 | **23.0** |
| $7 \times 7$ | 26.4 | **27.2** | 21.8 | **22.8** |
| $9 \times 9$ | 26.3 | **27.1** | 20.8 | **22.8** |

Table 5: Transfer evaluation on VOC 2007 test set from COCO-full dataset

| Method | Baseline | MetaAnchor | Search |
|:---:|:---:|:---:|:---:|
| $mAP@0.5(\%)$ | 82.5 | 83.1 | **83.3** |

### 4.1.3 Cross evaluation between datasets of different distributions

Though domain adaption or transfer learning [29] is out of the design purpose of MetaAnchor, recently the technique of weight prediction[10], which is also employed in the paper, has been successfully applied in those tasks [15, 14]. So, for MetaAnchor it is interesting to evaluate whether it is able to bridge the distribution gap between two dataset. More specifically, what about the performance if the detection model is trained with another dataset which has the same class labels but different distributions of object box sizes?

We perform the experiment on COCO-mini, in which we "drop" some boxes in the training set. However, it seems nontrivial to directly erase the objects in image; instead, during training, once we use an ground truth box which falls in a certain range (in our experiment the range is $\{(h, w)|50 < \sqrt{hw} < 100, -1 < \log \frac{w}{h} < 1\}$, around $1/6$ of the whole boxes), we manually assign the corresponding loss to 0. As for test, we use all the data in the validation set. Therefore, the distributions of the boxes we used in training and test are very different. Table 4 shows the evaluation results. Obviously after some ground truth boxes are erased, all the scores drop significantly; however, compared with the *RetinaNet* baseline, MetaAnchor suffers from smaller degradations and generates much better predictions, which shows the potential on the transfer tasks.

In addition, we train models only with COCO-full dataset and evaluate the transfer performace on VOC2007 test set [6]. We use two models: Baseline(RetianNet) and MetaAnchor, which achieve the best performace on COCO-full dataset with different architectures. In this experiment, we achieve 83.3% mAP on VOC 2007 test set, with 0.8% improvement in mAP compared with Baseline and 0.2% better than MetaAnchor, as shown in Table 5. Therefore, MetaAnchor shows a better tansfer ability than the RetinaNet baseline on this task. Note that the result is evaluated without sofa class, because there is no sofa annotation in COCO.

### 4.1.4 Data-independent vs. data-dependent anchor function generators

In Sec. 3.2 we introduce two variants of anchor function generators: data-independent (Equ. 4) and data-dependent (Equ. 6). In the above subsections we mainly evaluate the data-independent ones. Table 6 compares the performance of the two alternatives. For simplicity, we use the same training and test anchor configurations; the IoU threshold is 0.6/0.5. Results shows that in most cases data-dependent variant is slight better, however, the difference is small. We also report the scores after anchor configuration search (described in Sec. 4.1.2).

## 4.2 Results on COCO Object Detection

Finally, we compare our fully-equipped MetaAnchor models with *RetinaNet* [23] baselines on COCO-full dataset (also called *trainval35k* in [23]). As mentioned at the begin of Sec. 4, we follow the same evaluation protocol as [23]. The input resolution is $600\times$ in both training and test. The backbone feature extractor is *ResNet-50* [13]. Performances are benchmarked with COCO standard *mmAP* in the *minival* dataset.

Table 6: Comparison of anchor function generators (mmAP, %)

| # of Anchors | Data-independent | Data-dependent |
|:---:|:---:|:---:|
| $3 \times 3$ | 26.0 | **26.5** |
| $5 \times 5$ | **27.3** | **27.3** |
| $7 \times 7$ | 27.0 | **27.4** |
| $9 \times 9$ | **27.4** | 27.3 |
| search[3] | 27.6 | **28.0** |

Table 7: Results of YOLOv2 on COCO *minival* (%)

| Method | Baseline | MetaAnchor | Search |
|:---:|:---:|:---:|:---:|
| $mmAP$ | 18.9 | **21.2** | **21.2** |
| $mAP@0.5$ | 35.2 | 39.4 | **39.5** |

Table 8 lists the results. Interestingly, our reimplemented RetinaNet model is 1.8% better than the counterpart reported in [23]. For better understanding, we further investigate a lot of anchor box configurations (including those in Table 1) and retrain the baseline model, the best of which is named "RetinaNet\*" and marked with "search" in Table 8. In comparison, our MetaAnchor model achieves **37.5%** mmAP on COCO *minival*, which is $1.7\%$ better than the original RetinaNet (our implemented) and $0.6\%$ better than the best searched entry of RetinaNet. Our data-dependent variant (Equ. 6) further boosts the performance by $0.4\%$. In addition, we argue that for MetaAnchor the configuration for inference could be easily obtained by greedy search introduced in 4.1.2 without retraining. Specifically, the scales and aspects of greedy search anchors are $\{2^{k/5}| -2 < k < 6\}$ and $\{1/3, 1/t, 1, t, 3 | t = 1.1, 1.2, ..., 2\}$ respectively. Fig 2 visualizes some detection results predicted by MetaAnchor. It is clear that the shapes of detected boxes vary according to the customized anchor box $b_i$.

We also evaluate our method on PASCAL VOC 2007 and get preliminary resluts that MetaAnchor achieves $\sim 0.3\%$ more mAP than RetinaNet baseline (80.3->80.6% mAP@0.5). The gain is less significant compared with that on COCO, as we find the distribution of boxes on PASCAL VOC is much simpler than COCO.

To validate our method further, we implement MetaAnchor on YOLOv2 [31], which also use a two-layer network to predict detector parameters. For YOLOv2 baseline, we use anchors showed on open source project[4] to detect objects. In MetaAnchor, the "standard box" $(AH, AW)$ is (4.18, 4.69). For training, we follow the strategy used in [31] and use the COCO-full dataset. For the results, we report mmAP and mAP@0.5 on COCO *minival*. Table 7 illustrates the results. Obviously, MetaAnchor is better than YOLOv2 baseline and boosts the performace with greedy search method.

## 5   Conclusion

We propose a novel and flexible anchor mechanism named MetaAnchor for object detection frameworks, in which anchor functions could be dynamically generated from the arbitrary customized prior boxes. Thanks to weight prediction, MetaAnchor is able to work with most of the anchor-based object detection systems such as RetinaNet. Compared with the predefined anchor scheme, we empirically find that MetaAnchor is more robust to anchor settings and bounding box distributions; in addition, it also shows the potential on transfer tasks. Our experiment on COCO detection task shows that MetaAnchor consistently outperforms the counterparts in various scenarios.

**Acknowledgments**   This work is supported by National Key R&D Program No. 2017YFA0700800, China.

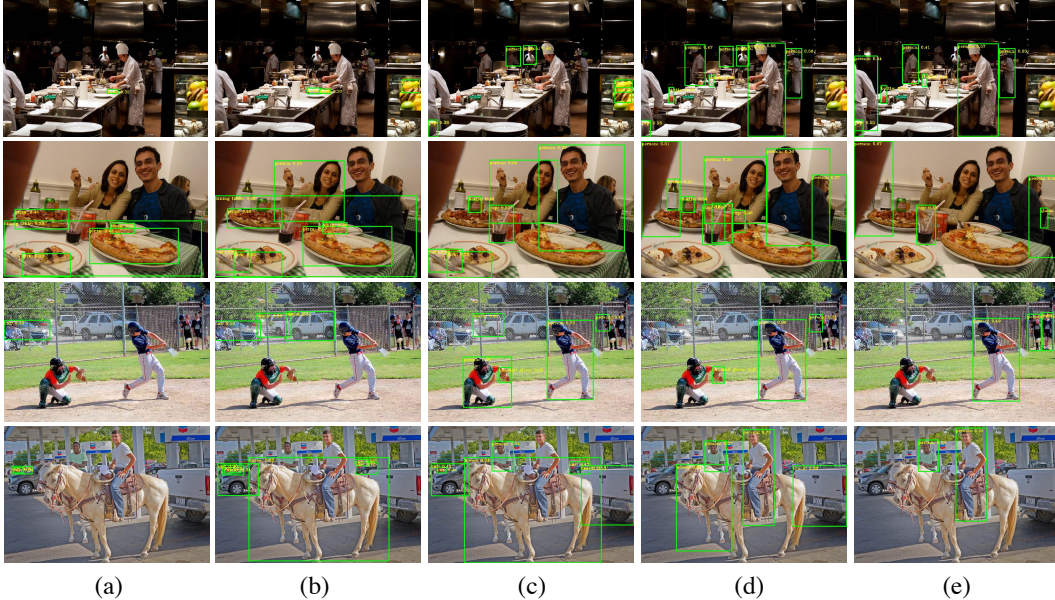

|  (a)  |  (b)  |  (c)  |  (d)  |  (e)  |

Figure 2: Detection results at a variety of customized anchor boxes. From (a) to (e) the anchor box sizes (scale, ratio) are: $(2^0, 1/3)$, $(2^0, 1/2)$, $(2^0, 1)$, $(2^0, 2)$ and $(2^0, 3)$ respectively. Note that for each picture we aggregate the predictions of all the 5 levels of detection heads, so the differences of boxes mainly lie in aspect ratios.

Table 8: Results on COCO *minival*

| Model | Training | Inference | |
|---|---|---|---|
| | # of Anchors | # of Anchors | mmAP (%) |
| RetinaNet [23] | $3 \times 3$ | $3 \times 3$ | 34.0 |
| RetinaNet (*our impl.*) | $3 \times 3$ | $3 \times 3$ | 35.8 |
| RetinaNet* (*our impl.*) | search | search | 36.9 |
| MetaAnchor (*ours*) | $3 \times 3$ | $3 \times 3$ | 36.8 |
| MetaAnchor (*ours*) | $9 \times 9$ | search | **37.5** |
| MetaAnchor (*ours*, data-dependent) | $9 \times 9$ | search | **37.9** |

## Footnotes

[1]There are a few recent studies on the topic, such as [33, 37].

[2]Here we follow the same definition of *scale* and *aspect ratio* as in [23].

[3]Based on the models with $7 \times 7$ anchor configuration in training.

[4]https://github.com/pjreddie/darknet

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
