[Reviews · NeurIPS 2018]

Reviewer 1



Summary: This paper proposes MetaAnchor, which is a anchor mechanism for object detection. In MetaAnchor, anchor functions are dynamically generated from anchor box bi, which describes the common properties of object boxes associated with i_th bin. It introduces a anchor function generator which maps any bounding box prior bi to the corresponding anchor function. In this paper, the anchor function generator is modeled as two-layer network for residual term R, added to the shared and learnable parameters for the anchor function theta^*. The residual term R can also depends on input feature x, which introduces the data-dependent variant of anchor function generator. Using weight prediction mechanism, anchor function generator could be implemented and embedded into existing object detection frameworks for joint optimization. Experiments to apply MetaAnchor to state-of-the-art single-stage detector RetinaNet on COCO dataset shows that MetaAnchor outperforms consistently outperforms the counterparts in various scenarios. Strengths: - The technical content of the paper appears to be correct with support of some experiment results. - This paper is clearly written with many details description and discussion on how optimization works and how experiment is performed. - There is some novelty of using neural network to learn anchor function. Weaknesses: - As mentioned in section 3.2, MetaAnchor does not show significant improvement for two-stage anchor-based object detection. - The experiment evaluation is only done for one method and on one dataset. It is not very convincing that MetaAnchor is able to work with most of the anchor-based object detection system. Rebuttal Response: Overall, I think this approach has value on one-stage frameworks. The experiment is only on COCO, but performed in an extensive way. It would be great to have more consistently good results on another benchmark dataset such as OpenImage. But for this submission now, I am willing the upscale my score from 4 to 6.

Reviewer 2



This paper provides a way for detection methods to get rid of the classic fixed set of predefined anchors which are always hard to choose. Implementing detection boxes as a function of continuous parameters representing the prior/anchors (instead of finite set of fixed prior) allows users to choose at inference time which prior/anchor they want to use. In addition they provide a system to predict the prior from the image so that the user's a priori is removed from the detection process by jointly optimizing both prior (which are outputs of a neural net) and the more classic part regression and scoring, which act on continuous inputs. Strengths: the approach is original, gives improvements over strong baselines. In addition, the approach is of general interest for the object detection community Weaknesses: The mathematical notations employed for the anchors and boxes are misleading, and the article would benefit from a user-friendly rewrite. Publishing at NIPS does not require to use overly complicated formalism. Under its current form, the paper is barely readable. I think the approach would be even more interesting on use cases where modalities are very different between train and test. For instance, training on large objects and testing on very small objects. Or even synthetic to real settings as proposed by the authors. Fixed anchors should fail in this setting. Generating data dependant anchors should provide big improvements it would be interesting to see.

Reviewer 3



The paper presents an anchor generation algorithm that learns anchor functions dynamically from arbitrary customized prior boxes rather than making them pre-defined. Results show improvements over a pre-defined anchor scheme on COCO detection task. Pros: + A simple algorithm and well motivated. Pre-defining anchor scales and aspect ratios is a tricky problem in object detection, making it learnable inside the deep network is a clear contribution. + The anchor generator is flexible and can theoretically be integrated into many object detection networks. + Reasonable improvements over the baseline. Cons: - The paper is only compared with [21] which uses a very simple strategy to design the anchor config. It's unclear how does the method compare with more carefully designed anchor strategies, e.g. via clustering (YOLO v2). - I think one clear advantage of the learned anchor configuration is through domain adaptation (i.e. train and test on different datasets). The experiments on cross evaluation between datasets is a bit trivial. It will be interesting to see experiments on actually different datasets such as Pascal and COCO. - It will be interesting to see how well does the proposed method work in another type of architecture, e.g. two-stage network.